# A New Approach of 3D Lightning Location Based on Pearson Correlation Combined with Empirical Mode Decomposition

**Yanhui Wang** [1,2,3,*]**, Yingchang Min** [1,2]**, Yali Liu** [1,2] **and Guo Zhao** [3]

1   Key Laboratory of Meteorological Disaster, Ministry of Education (KLME)/Joint International Research Laboratory of Climate and Environment Change (ILCEC)/Collaborative Innovation Center on Forecast and Evaluation of Meteorological Disasters (CIC-FEMD)/Key Laboratory for Aerosol Cloud-Precipitation of China Meteorological Administration, Nanjing University of Information Science & Technology, Nanjing 210044, China; 20201249010@nuist.edu.cn (Y.M.); 20201249007@nuist.edu.cn (Y.L.)
2   Key Laboratory of Middle Atmosphere and Global Environment Observation (LAGEO), Institute of Atmospheric Physics, Chinese Academy of Sciences, Beijing 100029, China
3   Public Technical Service Center, Northwest Institute of Eco-Environmental Resources, Chinese Academy of Sciences, Lanzhou 730000, China; guozh@lzb.ac.cn
*   Correspondence: wangyanh@nuist.edu.cn; Tel.: +86-15186827356

**Abstract:** To improve the accuracy of pulse matching and the mapping quality of lightning discharges, the Pearson correlation method combined with empirical mode decomposition (EMD) is introduced for discharge electric field pulse matching. This paper uses the new method to locate the lightning channels of an intra-cloud (IC) lightning flash and a cloud-to-ground (CG) lightning flash and analyzes the location results for the two lightning flashes. The results show that this method has a good performance in lightning location. Compared with the pulse-peak feature matching method, the positioning results of the new method are significantly improved, which is mainly due to the much larger number of positioning points (matched pulses). The number of located radiation sources has increased by nearly a factor of seven, which can significantly improve the continuity of the lightning channel and clearly distinguish the developmental characteristics. In the CG flash, there were three negative recoil streamers in the positive leader channel. After the three negative recoil streamers were finished, taking approximately 1 ms, 12 ms, and 2 ms, respectively, the negative leader channel underwent a *K*-process. The three negative recoil streamers are not connected to the *K*-processes in the negative leader channel. We think that the three negative recoil streamers may have triggered the three *K*-processes, respectively.

**Keywords:** lightning 3D location; EMD; Pearson correlation; pulse matching; negative recoil streamer

## 1. Introduction

Electromagnetic radiation emitted during lightning breakdown, and the location and structure of lightning can be mapped out by locating multiple radiation sources. The application of modern signal processing technology and high-speed data acquisition technology has improved the ability and level of lightning detection. Lightning radiation source positioning technology can help to realize the analysis of the temporal and spatial evolution process of lightning channel, which has great significance for the study of lightning physical mechanisms and lightning protection. Interferometric mapping [1] is one method of lightning location, and the time-of-arrival (TOA) technique is another major method for lightning location. In recent decades, the TOA technique [2–10] has been widely used to locate lightning radiation sources. The TOA method is roughly divided into four steps, namely, signal acquisition, data preprocessing, pulse extraction and pulse matching, and solving the location and occurrence time of the radiation source. Each step is critical to generating lightning location results.

For pulse extraction, Lyu et al. [11] extracted all pulses greater than the noise level and obtained the arrival time of each pulse. This method for measuring the peak time

of individual pulses is similar to the approaches in previous studies [12–14]. However, this method has two problems. The first problem is that the low frequency may raise the amplitude of some interference pulses, causing them to be misidentified. The second problem is that a bipolar pulse may be identified as two pulses.

For pulse matching, the broadband cross-correlation method was applied by Qiu et al. [15] to map an artificially triggered lightning flash. Then, Cao et al. [16] and Sun et al. [17–19] applied a general correlation time-delay estimation algorithm based on the broadband cross-correlation method and wavelet transformation to realize the pulse matching of the same discharge event. There is another way to match pulses. The 3D lightning radiation source location system [20–23] and Shi et al. [24] obtained the normalized power waveform after Hilbert transformation of the original waveform; then, they extracted the peak information of the pulse from the normalized power waveform. Finally, the pulse-peak feature matching method is used to finish pulse matching. If the arrival time difference between two stations is less than the optical path difference between the two stations (ratio of distance to speed of light), then the matching of the two pulses is reasonable. To improve the accuracy of pulse matching, Fan et al. [8] applied empirical mode decomposition (EMD) to lightning signal processing to achieve low-frequency filtering and high-frequency noise reduction of multi-station lightning electric field waveforms. For the TOA method of lightning positioning, the EMD method has better results than the Hilbert transform method in data processing.

However, matching a pair of pulses with similar amplitudes will also lead to the situation in which partial discharge events cannot be effectively matched, so the method needs to be further improved. Due to the inherent mode aliasing phenomenon of EMD, after the EMD method is used to process the lightning electric field signal, there are still multi-station waveform mismatches, thereby resulting in inaccurate pulse matching by using a cross-correlation algorithm [9].

To improve the accuracy of pulse matching and the mapping quality of lightning discharges, the Pearson correlation method combined with the EMD method is introduced for discharge electric field pulse matching. The EMD method is used to decompose the original signal into multiple components; then, a new signal is obtained by removing the residual component and superimposing other components in the same time domain. After using EMD to process the lightning broadband electric field signal, Pearson correlation is applied to pulse matching. In order to further verify the better performance of this new method, this paper uses this new method to locate the lightning channels of an IC lightning flash and a CG lightning flash. In addition, the positioning results are analyzed and compared with the positioning results of the pulse peak feature matching method.

## 2. Experiment and Equipment

The 3D lightning radiation source location system [20–23] is established in Datong County, Qinghai Province, China. The data used and analyzed in this paper come from the lightning broadband electric field change measurement network of this system. The Datong region is located in the northeastern part of Qinghai Province, with an average altitude of approximately 2600 m. It is a mountainous area with complex terrain and obvious altitudinal differences in climate. Due to the high altitude of the area, its climate is affected by the interaction between the plateau weather system and the westerly belt weather system. The complexity of the topography of the river valley and the nature of the underlying surface results in more short-term heavy rainfall. Moreover, the region often receives strong convective weather, such as thunderstorms and hail [25–32].

Figure 1 shows the distribution of the seven stations of the 3D lightning radiation source location system. With station MD as the center, seven stations are distributed in an area with a radius of approximately 8 km. The longitude, latitude, and altitude of the station MD are 101.6200592° E, 37.0133483° N, and 2493.02 m, respectively. Each station of the network was mainly composed of a very high-frequency (VHF) antenna, broadband electric field change measurement antenna (10 MHz bandwidth and 100 μs time

constant), band-pass filter, logarithmic amplifier, high-speed A/D data acquisition card, high-precision clock (time precision of 50 ns), processor, and wireless data transmission system. In addition to this equipment, the station MD (the main station) was equipped with a VHF narrowband interferometer. The sampling rate of the high-speed A/D data acquisition card is 20 MS/s, and the bandwidth of the broadband electric field system is 0–10 MHz. To amplify the received electric field signal without distortion, the system uses a logarithmic power amplifier circuit. The peak amplitude and peak time of the lightning radiation signal are recorded and buffered by the digital module. The data recording length was determined by the baseline length and the resolution of the processed data. The single data recording time used in this paper was 1.2 s, and the system triggering and data recording times were determined with a GPS-synchronized high-precision clock. The noise level of the signal was controlled by a synchronous trigger threshold circuit. The received signal was sent to the station MD through the wireless data transmission system in real time for time difference calculation, real-time processing, and display for lightning radiation source location.

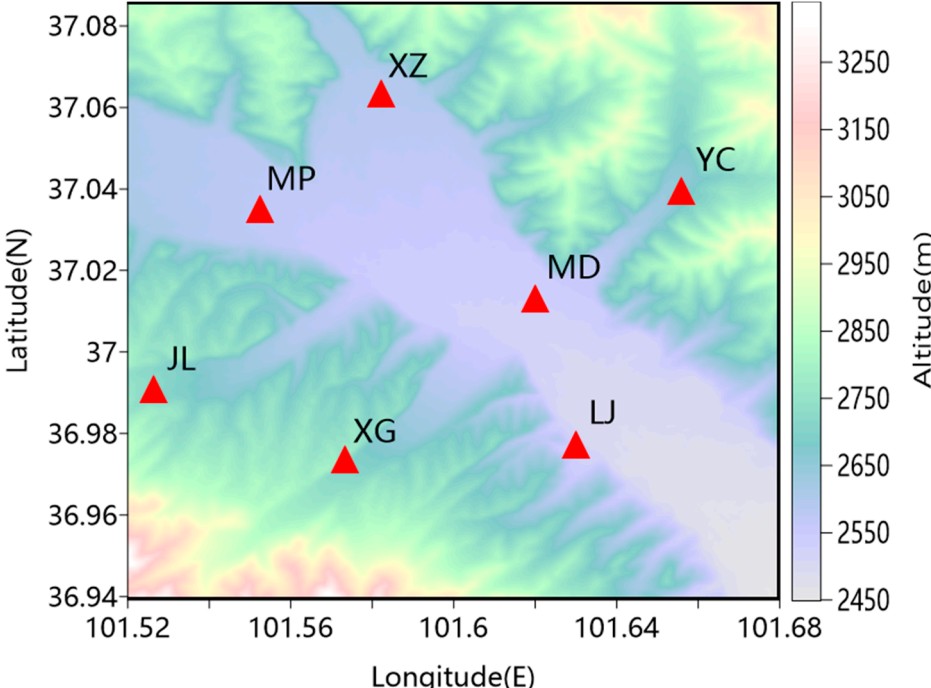

**Figure 1.** The seven stations of the 3D lightning radiation source location system in the Datong region of Qinghai Province, China.

## 3. Method

The time-of-arrival (TOA) technique was used to locate the lightning radiation sources. For the same lightning radiation source, $(x_i, y_i, z_i)$ and $t_i$ represent the three-dimensional coordinates of station $i$ and the pulse arrival time, respectively. $(x, y, z)$ and $t$ represent the spatial location and occurrence time of a radiation source, respectively, and $c$ is the propagation speed of electromagnetic waves in the air ($3 \times 10^8$ m s$^{-1}$). The arrival time of the radiation pulse at station $i$ satisfies Equation (1) as follows:

$$t_i = t + \frac{1}{c}\sqrt{(x_i - x)^2 + (y_i - y)^2 + (z_i - z)^2}. \tag{1}$$

Through the TOA method, the location and occurrence time of a lightning radiation source is calculated, and the complete lightning channel can be depicted by multiple calculated radiation sources. Solving for the values of $x$, $y$, $z$, and $t$ should minimize the chi-square value $x^2$, which can be expressed as Equation (2). The minimum chi-square

value $x^2$ can be computed using the nonlinear least squares method, which was introduced in [10,12]. First, select at least four stations from all stations to calculate the position of the radiation source, and keep trying different combinations of stations to minimize the chi-square value $x^2$. The $i$ in Equation (2) represents the $i$-th station, and $N$ is the number of stations in the lightning detection network. $t^{obs}$ is the pulse arrival time, and $t^{fit}$ is the fitted pulse arrival time. $\Delta t_{rms}$ is the time error, which is determined by the performance of the hardware system. Finally, all the positioning points are screened by chi-square value ($x^2$), under the condition of $x^2 < 5$.

$$x^2 = \sum_{i=1}^{N} \frac{\left(t_i^{obs} - t_i^{fit}\right)^2}{\Delta t_{rms}^2} \tag{2}$$

*3.1. Data Preprocessing*

Empirical mode decomposition (EMD) is a method for processing nonlinear and non-stationary signals. EMD can decompose complex signals into a finite number of intrinsic mode functions (IMFs). Details of the specific principle and algorithm of EMD are found in [8,33]. To decompose the intrinsic mode functions from the original signal, the process of the EMD method used in this paper is expressed as follows. For all the extreme points of the original signal $X_i(t)$, the envelopes of the upper and lower extreme points are fitted with a tertiary spline curve. The original signal is subtracted from the average of the two envelopes to obtain $c_1$. It is judged according to the preset criterion whether $c_1$ is an IMF. If not, $X_i(t)$ is replaced with $c_1$, and the above steps are repeated until $c_k$ meets the criterion (we assume in advance that this sifting process has been repeated $k$ times).

In the EMD decomposition process, each pre-extracted component must be sifted and judged, whether it is the IMF component that we need. For the IMF component, all the local maxima are positive, and all the local minima are negative. Sift relative tolerance (it was set as 0.2, referring to [8,33]) is one of the sifting stop criteria; that is, sifting stops when the current relative tolerance is less than the sift relative tolerance.

It should be noted that in practice, the average value of the upper and lower envelopes is usually not 0, so a stop criterion of the sifting process is set [8], that is:

$$SD_k = \sum_{t=0}^{T} \frac{|c_{k-1}(t) - c_k(t)|^2}{c_{k-1}^2(t)} \leq \varepsilon. \tag{3}$$

$\varepsilon$ generally takes values between 0.2 and 0.3 [8,33], and $SD_k$ is the current relative tolerance of the $k$-th sifting of an IMF component. We can repeat this sifting process $k$ times and extract an IMF component when the relative tolerance is less than 0.2.

We subtract this component from the current signal and repeat the above steps to get the next component. The signal for the subsequent processing is obtained by subtracting all the previously obtained IMF components from the original signal. In this paper, the decomposition stopped because the max number of IMFs was reached (the max number of IMFs is generally 10). After the EMD, decomposition stops, and the remaining signal $r_n$ is the residual component. The residual component $r_n$ usually contains direct current (DC) components, monotonic components, and very low-frequency (VLF) components. Using the EMD method to decompose the original signal $X_i(t)$ into a series of IMF components and the residual component $r_n$ can be expressed as:

$$X_i(t) = \sum_{i=1}^{10} \text{IMF}_i + r_n. \tag{4}$$

According to the signal decomposition principle of EMD, the original signal is gradually decomposed into components of different varying scales from high frequency to low frequency. Then, a new signal is obtained by removing the residual component and

superimposing other components, in the same time domain, which can be expressed as Equation (5).

$$x_i(t) = \sum_{i=1}^{10} \text{IMF}_i \tag{5}$$

It is necessary to subtract the residual component from the original data. The amplitude of the residual component changes more slowly with time than the radiation pulse, which are not caused by the air breakdown of lightning. Removal of the residual component can reduce the low-frequency interference caused by the signal acquisition system. This is one advantage of EMD when dealing with nonlinear and non-stationary signals. The VLF signals are usually used to locate the channel of the return stroke of a CG flash. However, this work is dedicated to the refined positioning of lightning channels. It is necessary to extract and match more small pulses (spikes). The existence of VLF signals will affect the pulse extraction and pulse matching on the microsecond time scale. The removal of the residual component helps to identify weaker pulses as well as identify bipolar pulses and extract their main spike.

Normalization is a way to simplify calculations; that is, a dimensional expression is transformed into a dimensionless expression, which becomes scalar, and the signal is limited to a specific range. Normalization is for the convenience of subsequent pulse matching. The distance between each station and the lightning radiation source is different. The signal amplitude of some stations is small due to signal propagation attenuation.

After removing the residual component from the original signal of the *i*-th station, the signal $x_i(t)$ is obtained, and the signal $y_i(t)$ is obtained by normalization. Normalization can be expressed as Equation (6). Among them, $\max(x_i)$ is the maximum value of signal $x_i(t)$, and $\min(x_i)$ is the minimum value of signal $x_i(t)$. After the normalization process is completed, the difference between the maximum value and the minimum value of each station signal is 2, and the pulse amplitudes between different stations are similar, which is important to achieve pulse matching through Pearson correlation.

$$y_i(t) = \frac{2(x_i(t) - \min(x_i))}{\max(x_i) - \min(x_i)} - 1 \tag{6}$$

### 3.2. Discharge Electric Pulse Matching

The old method (pulse-peak feature matching) is based on two features of peak amplitude and arrival time for pulse matching. First, extract the pulses from all stations and determine whether the arrival time difference is less than the optical path difference between the two stations. On this basis, find the pulse with the closest amplitude to finally achieve pulse matching. This method may incorrectly match pulses due to several pulses with similar amplitude in the period of optical path difference.

To improve the quality of lightning-refined positioning, this study applies the Pearson correlation method combined with the EMD method to achieve accurate pulse extraction and pulse matching. The mathematical description of the Pearson correlation coefficient is as follows:

$$\rho(A, B) = \frac{\text{cov}(A, B)}{\sigma_A \sigma_B} = \frac{\sum\limits_{n}(A_n - \overline{A})(B_n - \overline{B})}{\sqrt{(\sum\limits_{n}(A_n - \overline{A}))(\sum\limits_{n}(B_n - \overline{B}))}}. \tag{7}$$

Covariance (*cov*) is an indicator that reflects the correlation degree between two random variables. If one variable becomes larger or smaller at the same time as another variable, then the covariance of the two variables is positive. $\sigma_A$ is the variance of dataset $A$ (a 25 μs time series in this paper). $\overline{A}$ is the average value of dataset $A$, and $\overline{B}$ is the average value of dataset $B$.

It should be pointed out that the Pearson correlation coefficient has an obvious shortcoming: when the number of samples $n$ is small, the correlation coefficient fluctuates greatly, and the absolute value of the correlation coefficient is easily close to 1; however,

when $n$ is large, the absolute value of the correlation coefficient is likely to be small. The number of samples $n$ in Equation (7) is set to be 500 (consider the data sampling rate and the number of samples), and it can be adjusted appropriately according to the data sampling rate.

The Pearson correlation is similar to the traditional cross-correlation method. The most similar waveform is found through the shift between the time series, but there are some differences. The Pearson correlation coefficient can measure the similarity of two datasets, and the output range is $-1$ to $+1$. Among them, 0 means no correlation, positive value means positive correlation, and negative value means negative correlation, which has long been used in many fields such as mathematical statistics. If the changes of the two signals are almost synchronized, even if the pulse amplitude is very small, a Pearson correlation coefficient close to 1 can still be obtained, which obviously exceeds the similarity threshold. However, for the traditional cross-correlation method, even if the signal synchronization is very good, only a small value of the cross-correlation function can be obtained. Such a small value of the cross-correlation function may be caused by interference. It is difficult to judge which situation is correct. Of course, the traditional cross-correlation function has a good performance in LF/VLF lightning location; usually, only those larger pulses need to be located to achieve long-distance lightning location.

The application of Pearson correlation still requires determining the time window. The time window of pulse matching in this paper is 75 µs (1500 data points). The signal arrival time difference of the same discharge event between different stations is less than or equal to the ratio of the linear distance between the two stations to the signal propagation speed (speed of light). This ratio is called the optical path difference. The length of the time window is at least twice the optical path difference from the main station (station MD) to the sub-station. The longest linear distance from the main station to the six sub-stations in this lightning location network is 8687 m, and the optical path difference is 28.96 µs. Therefore, the length of the time window is set as 75 µs. For the pulse matching of the same discharge event, the 75 µs time window of the six sub-stations is synchronized with the GPS time of the station MD. A time series (25 µs) is taken from the station MD and centered on the single pulse peak. Then, we found the most similar time series (25 µs) within the time window of these six sub-stations.

The result of pulse matching is to obtain the peak time in the middle of the most similar time series (25 µs) within the time window (75 µs). Where pulses are abundant, the time windows will overlap partially rather than completely. At most, one set of pulses can be obtained for each Pearson correlation matching, and the same set of pulses will not be used repeatedly.

In a time window of the electric field waveform, the following relationship must be satisfied for pulse matching:

Conditions (1) $\rho_0 > \alpha$
Conditions (2) $\rho_0 = \max(\rho)$

where $\alpha$ is the similarity threshold of the Pearson correlation coefficient ($\alpha$ is set to be 0.35 in this paper) and $\max(\rho)$ is the maximum value of the Pearson correlation coefficient of two stations in the time window. If conditions (1) and (2) are met, two time series between the station MD and a sub-station will be matched. Then, the time window is moved to the next MD's pulse to complete the pulse matching of other radiation sources.

## 4. Results of Data Processing and Pulse Matching

An intra-cloud (IC) lightning flash and a cloud-to-ground (CG) lightning flash were recorded from a nocturnal thunderstorm. By convention, the IC lightning flash is labeled as 001136 according to the time of occurrence, and the CG lightning flash is labeled as 000241. The time axis information in all the figures is unified. For example, Figure 2 shows the electric field waveforms of IC flash 001136; the time 37.1 in Figure 2 can be expressed as

00:11:37.1 in the time format of 'h-min-sec', and the time information of 'h-min' is omitted in these figures.

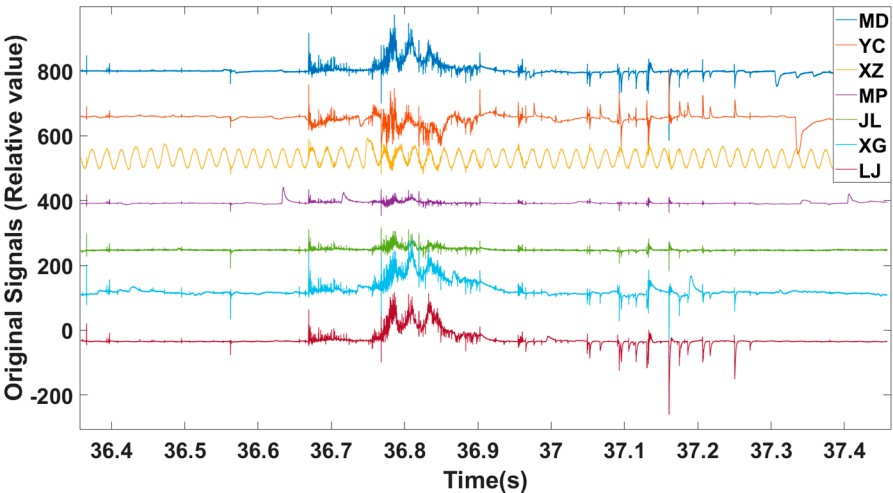

**Figure 2.** Original electric field signal over time from the seven stations for IC flash 001136.

### 4.1. Results of Data Preprocessing

The original electric field signal waveforms from the seven stations are shown in Figure 2. Figure 3 shows the EMD decomposition results of a lightning signal (IC flash 001136) at the MD station, which shows the waveforms of 10 IMF components and the residual component. In this paper, the original signal of each station is decomposed by the EMD method, and then, a new signal is obtained by removing the residual component and superimposing other components in the same time domain. Then, the synthesized signal are normalized, as shown in Figure 4.

Station XZ has 50 Hz oscillation interference, which can be resolved by removing the residual component. In the original signal shown in Figure 2, there are several large peaks in the part where the time is >37.3 sec., but these large peaks are removed in Figure 4. It should be emphasized that this change is a significant improvement, which is due to that EMD decomposing the original signal and removing the residual component. The signals of different change scales are gradually extracted from high frequency to low frequency, and finally, these slower electric field changes are included in the residual component.

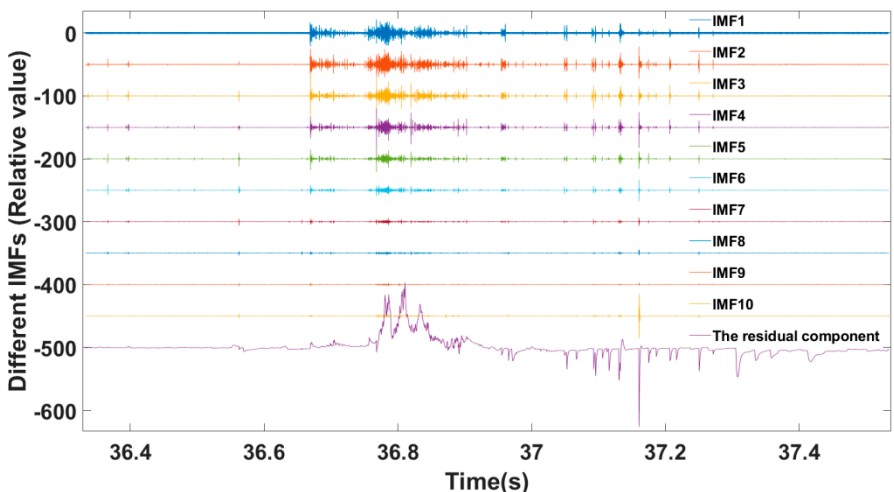

**Figure 3.** Different components of station MD obtained by the EMD decomposition method for IC flash 001136.

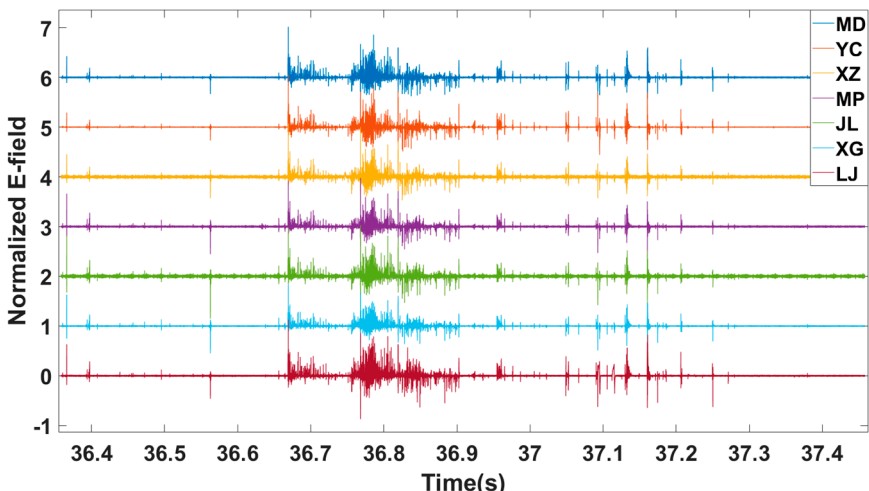

**Figure 4.** Normalized electric field signal over time from the seven stations for IC flash 001136.

*4.2. Results of Pulse Matching*

Figure 5 shows only the 9 ms normalized electric field waveform of IC flash 001136 detected by the station MD and the extracted pulses. Each pulse is emitted from an air breakdown, and a flash contains many air breakdowns in a short period of time. Each extracted pulse peak is marked by dots with the gradient color by time. Ninety-three lightning radiation pulses were extracted from this waveform, as shown in Figure 5.

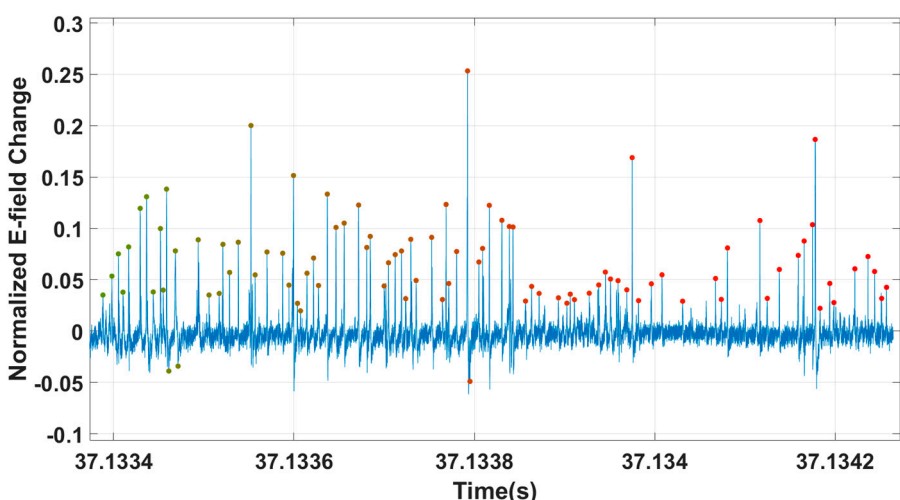

**Figure 5.** Part of the normalized electric field waveform of IC flash 001136 detected by the station MD and these 93 extracted pulses.

For the pulse extraction, if the amplitude of a certain spike relative to its left and right extreme values is greater than the noise threshold, and the peak-to-peak interval is greater than 2.5 microseconds, then the spike is identified as a pulse. This work only needs to identify the pulses of the station MD to help determine the position of each time window. Meanwhile, other stations need to find the most similar time series within the time window that satisfies the optical path difference and then further search for the extreme value in the middle of the time series as a matched pulse. The purpose of pulse matching is to accurately obtain the time of a single pulse spike. At the same time, it must be pointed out that the high-frequency electric field noise of the station MD is within the range of $\pm 8$ V/m, and the noise threshold is converted according to this value and scaled in the same proportion as the data normalization process.

After removing the residual component from the original signal, the signal is normalized. The tiny signal fluctuates near a straight line so that we can accurately extract more weaker pulses, and the number of extracted pulses is increased. At the same time, we are easy to identify bipolar pulse, to prevent the tail spike of the bipolar pulse from being identified as a single pulse, making the number of radiation sources more reasonable. As shown in the pulse extraction result in Figure 5, there are other spikes that are not marked as pulses (especially negative). These negative spikes are judged to be the tail spikes of bipolar pulses, and the tail spikes of these bipolar pulses are not needed. After data preprocessing, we can extract more small pulses and avoid a bipolar pulse from being mistakenly identified as two pulses.

Figure 6 shows a set of matched pulses received by the seven stations. This set of matched pulses is emitted from a same air breakdown and is marked with black dots. It can be seen that in the signal waveform with abundant pulses and uneven amplitude, the new method can accurately complete pulse matching.

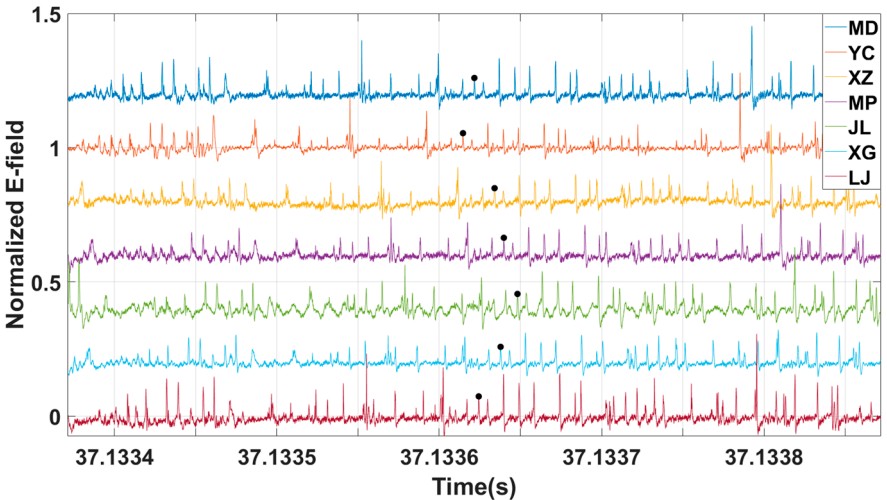

**Figure 6.** A set of matched pulses.

## 5. Positioning Results and Analysis of Lightning Physical Processes

### 5.1. Positioning Efficiency of the New Method

The GPS-based TOA technique has been proven with a high accuracy for locating lightning VHF radiation sources in the previous research results [4,34,35]. Based on the lightning pulse arrival time recorded by the GPS high-precision clock, a new method is proposed in this paper to achieve signal preprocessing and pulse matching, and it locates the lightning radiation sources through the TOA technique. Figure 7 shows the height and the number of radiation sources corresponding to the extracted pulses in Figure 5. Figures 8 and 9 are the positioning results of IC flash 001136 obtained by the old method and the new method, respectively. Figure 10 shows the 45° top view of IC flash 001136 from east to west. Figure 11 shows the positioning results of CG flash 000241 by the new method.

As shown in Figure 5, ninety-three lightning radiation pulses were extracted from this segment of MD waveform. Figure 7 shows the height of the radiation sources corresponding to the pulses in Figure 5. In total, ninety-two lightning radiation sources were located for the period using the new method. Obviously, the new method has a good performance in lightning location.

### 5.2. The 3D Location Results of Two Lightning Flashes

#### 5.2.1. IC Flash 001136

Using the algorithm of the 3D lightning radiation source location system [20–23] and the new method introduced in this paper to locate the IC lightning flash 001136, the results in Figures 8 and 9 are obtained, respectively. As shown in these two figures, the positioning

results of the radiation sources of IC lightning flash 001136 are colored by time and plotted with different projections. Each point represents the location of a breakdown event.

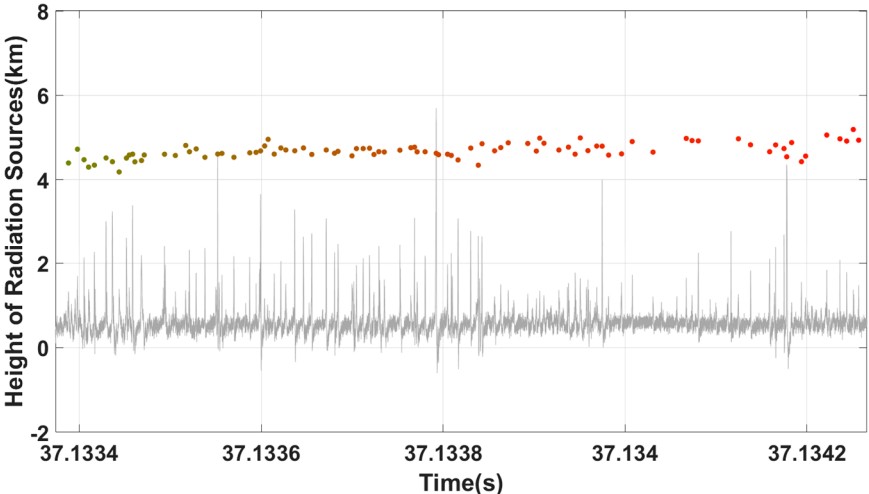

**Figure 7.** The height of the radiation sources corresponding to the pulses in Figure 5.

As shown in Figure 8, the old method located 568 radiation sources for IC flash 001136. Progressively, the new method located 3699 radiation sources for the same lightning flash, as shown in Figure 9. The number of located radiation sources has increased by nearly a factor of seven. Compared with the old method, the positioning results of the new method are significantly improved, which is mainly due to the much larger number of positioning points (matched pulses). The two main reasons for this significant improvement are the increase in the number of extracted pulses and the application of Pearson correlation. The reasons for the improvement in pulse extraction are explained in Section 4.2 above. Obviously, accurate pulse extraction and pulse matching can increase the number of positioning points and make the lightning channel clearer. The old method (pulse-peak feature matching) may incorrectly match pulses due to several pulses with similar amplitude in the period of optical path difference. The new method realizes pulse matching based on the similarity of time series (25 μs), and it can still obtain a high Pearson correlation coefficient in a time series with good synchronization but a small amplitude. Moreover, the same pulse will not be reused.

The extension of the north–south direction of this lightning reaches 14 km, but the previous method can only locate the scale of about 10 km, and the middle of the channel cannot be connected. After using the new method to achieve data preprocessing and pulse matching, the lightning channel is clearly and continuously located. It can be seen from the positioning results that the IC flash has a typical double-layer structure, in which the positive leader propagates in the lower negative charge region, and only a small amount of electromagnetic pulses emitted by it can be detected. At the same time, the negative leader propagates in the upper positive charge region and emits abundant electromagnetic pulses. The height in Figures 7, 8 and 10 only represents the vertical distance between the radiation source and the station MD.

From the 3D location results, the horizontal distance from the initiation point of IC flash 001136 to station MD was estimated to be approximately 9 km, with an initiation altitude of approximately 3 km. This IC flash started from a bidirectional leader, in which the positive leader developed downward into the lower positive charge region and then developed horizontally, while the negative leader developed upward into the positive charge region and then developed horizontally. This is a feature that the positive leader emits fewer radiation pulses during its development, while the negative leader emits abundant radiation pulses. Therefore, based on the feature, it is easy to distinguish the polarity of the leader and the polarity of the charge region. From Figure 9a, in the initial

stage of this lightning, it can be seen that the upward negative leader can be clearly located, while the positive leader can hardly be located within about 30 ms of the lightning beginning. Li et al. [36] also described a similar phenomenon through broadband VHF observations and believes that it may be because the higher power signal emitted by the negative leader covers the radiation signal of the developing positive leader, making it difficult to locate the positive leader. Note that most of the radiation sources occurred in the positive charge region existing between 3.5 and 6.0 km throughout this lightning, as shown in Figure 9c. The IC lightning flash extended the channel from the lightning start area to the south. Approximately 490 ms after the flash onset, a typical *K*-process occurred, and its duration was approximately 4 ms. The average velocity of the *K*-process discharge was approximately $1.25 \times 10^6$ m s$^{-1}$ and is a typical value of those reported by other researchers [6,37,38]. The whole flash lasted approximately 650 ms.

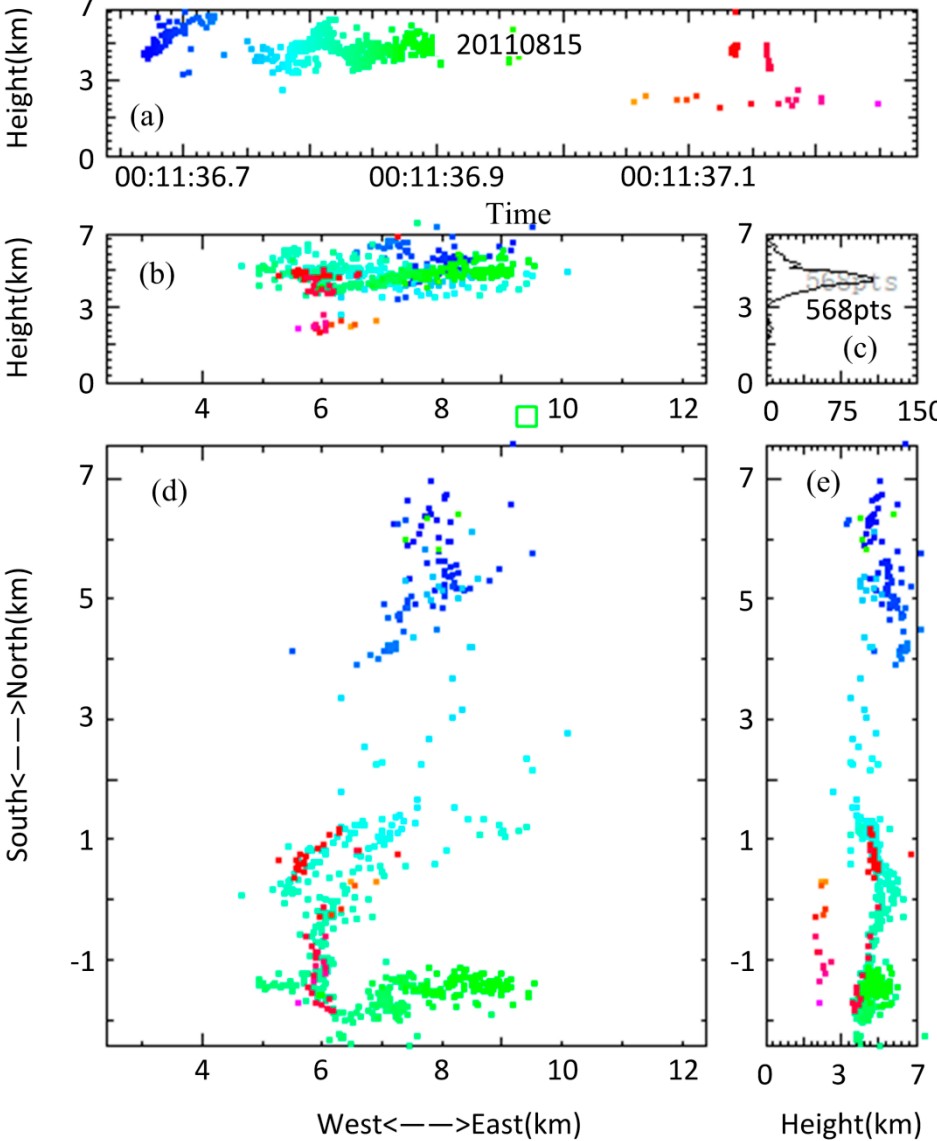

**Figure 8.** Location results for IC flash 001136 using the algorithm of the 3D lightning radiation source location system [20–23]. In the figure, (**a**) the height of the radiation source changes with time, (**b**) the vertical projection of the east-west direction, (**c**) the number of radiation sources changes with the height, (**d**) the horizontal projection, and (**e**) the vertical projection of the north-south direction.

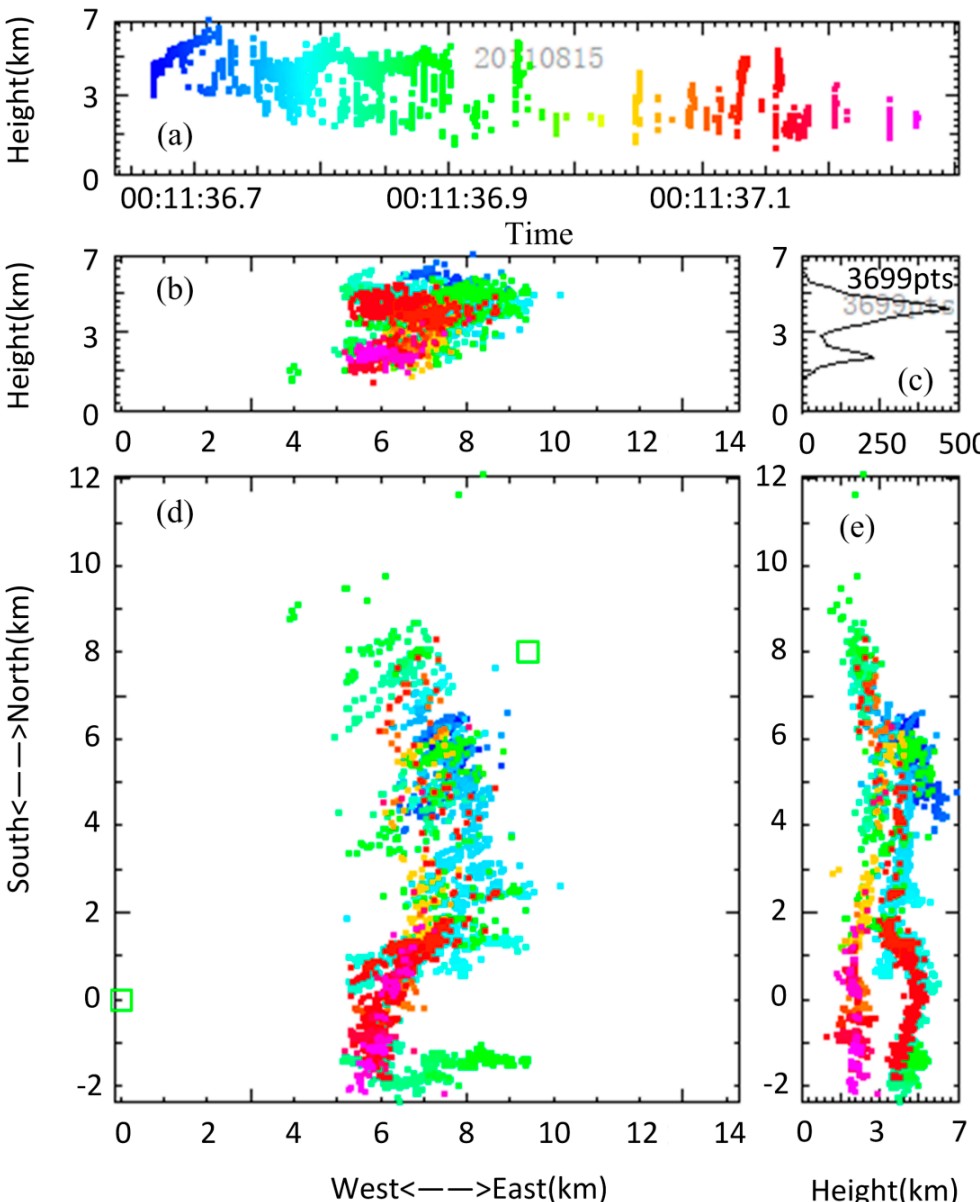

**Figure 9.** Location results for IC flash 001136 using the new method. In the figure, (**a**) the height of the radiation source changes with time, (**b**) the vertical projection of the east-west direction, (**c**) the number of radiation sources changes with the height, (**d**) the horizontal projection, and (**e**) the vertical projection of the north-south direction.

To make the structure of the lightning channel more intuitive, we provide a 45-degree angle top view to the horizontal of IC flash 001136 from east to west (Figure 10), which was obtained by the new method and algorithm introduced in this paper.

### 5.2.2. CG Flash 000241

Figure 11 presents the radiation source locations colored by time for CG flash 000241. In this paper, 5017 lightning radiation sources for CG flash 000241 have been located by the new method. From the 3D location results, the horizontal distance from the initiation point of CG flash 000241 to station MD was estimated to be approximately 9 km, with an initiation altitude of roughly 2–3 km.

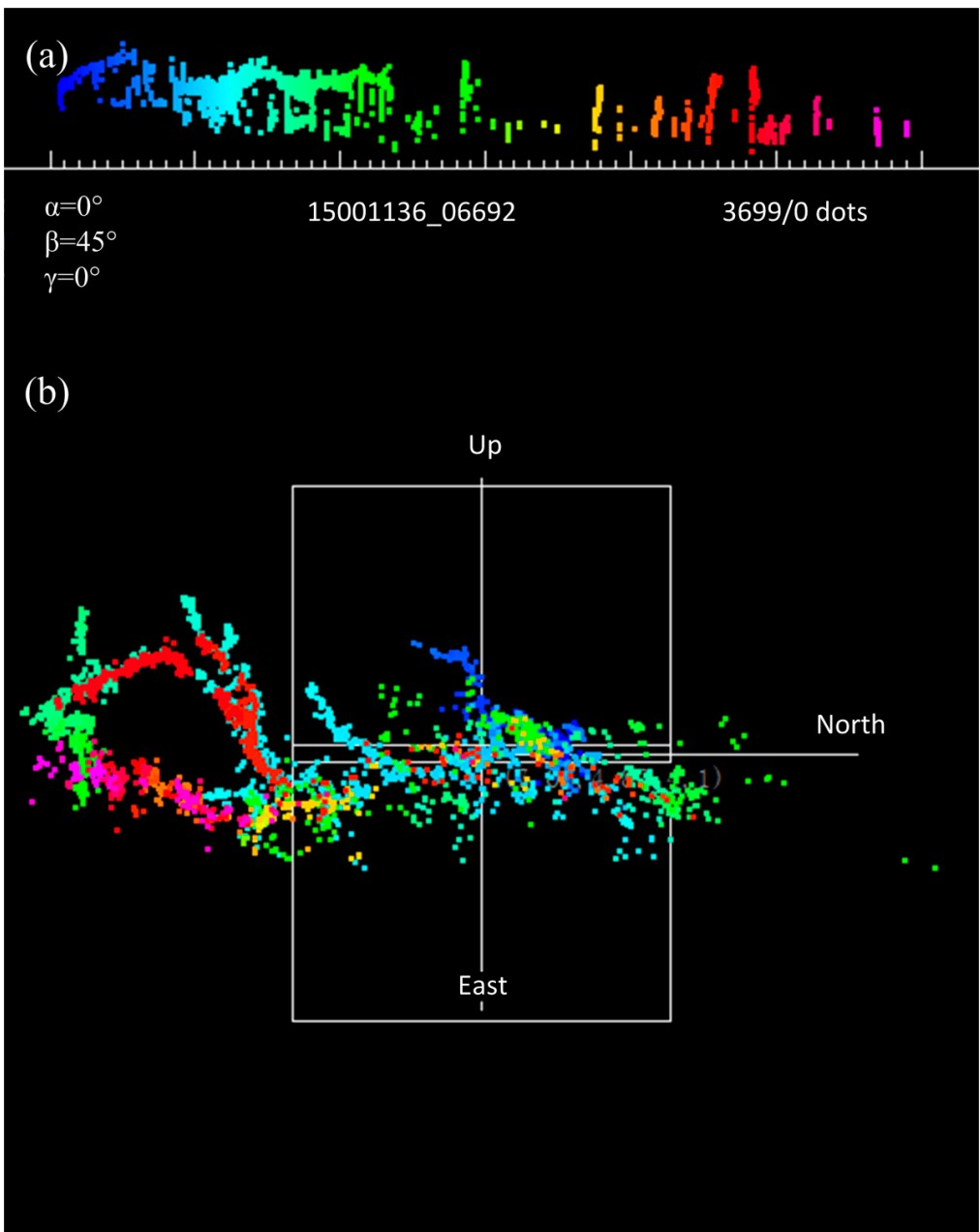

**Figure 10.** The 45° angle top view to the horizontal of IC flash 001136 from east to west. (**a**) Height-time plots and (**b**) the 45° top view from east to west.

The channel developed horizontally during the first 70 ms of the CG flash after initiation. Then, the lightning channel developed downwards for 40 ms to connect with the ground. Obviously, this is a single-stroke CG lightning flash, similar to lightning reported by previous studies [6,39]. The CG flash extended over a wide range to the northwest and southeast of the seven stations network. As shown in Figure 11, after the end of the single stroke, a negative leader developed horizontally from the lightning initiation region to the southeast, while a positive leader developed horizontally to the northwest.

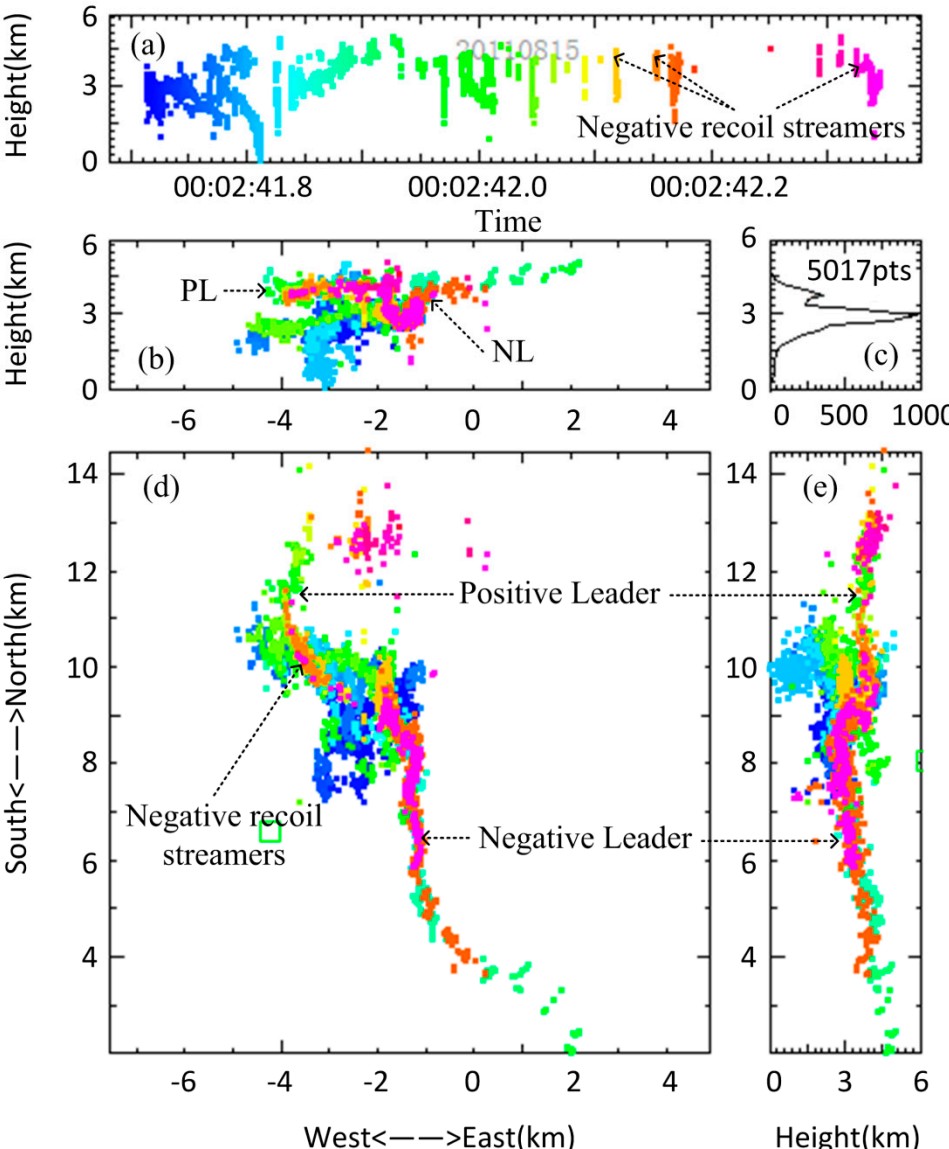

**Figure 11.** The VHF radiation sources of CG flash 000241. The different panels show (**a**) the height of the radiation source changes with time, (**b**) the vertical projection of the east-west direction, (**c**) the number of radiation sources changes with the height, (**d**) the horizontal projection, and (**e**) the vertical projection of the north-south direction.

The recoil streamer associated with the *K*-process propagates backwards from the tip of the positive leader channel to the lightning initiation region along the formed positive leader channel, and its average velocity is about $10^7$–$10^8$ m s$^{-1}$ [2,36,37,40–42]. From the dynamic positioning results of the CG flash, it is found that there were three negative recoil streamers in the positive leader channel. Figure 12 shows the pulses of the second negative recoil streamer and the second *K*-process pulses of the negative leader channel. The pulses amplitude of the second negative recoil streamer is smaller than the pulses of the second *K*-process of the negative leader channel, and the duration of the negative recoil streamer is very short (0.2 ms), with a horizontal length of approximately 3 km. The average velocity of the second negative recoil streamer was estimated to be approximately $1.5 \times 10^7$ m s$^{-1}$ and is typical of those reported by other researchers [2,37,41]. According to the positioning results of the CG lightning flash, after the three negative recoil streamers are finished (approximately 1 ms, 12 ms, and 2 ms), the negative leader channel undergoes a *K*-process. The negative recoil streamers are not connected to the *K*-processes. We think that the

three negative recoil streamers of the positive leader channel may have triggered the three *K*-processes of the horizontal negative leader channel of the lightning, respectively.

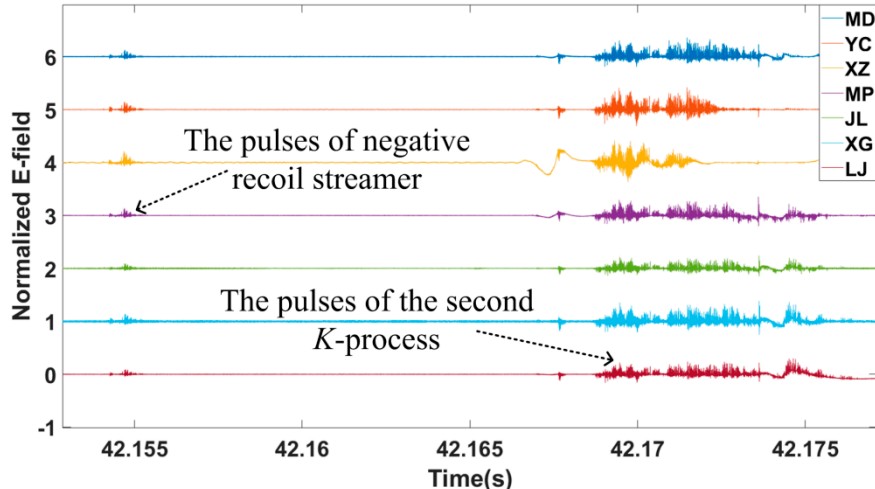

**Figure 12.** The pulses of the second negative recoil streamer and the second *K*-process pulses of the negative leader channel.

For the positioning results, the estimated error is that the horizontal error in the network is less than 60 m, and the vertical error is less than 180 m. This error (or location uncertainty) is obtained by the error estimation method of reference [4], and a more accurate positioning error needs to be determined by artificially triggered lightning experiments.

The original waveforms and the normalized waveforms of the CG flash are shown in Figures 13 and 14, respectively. As shown in the red box in Figures 13 and 14, it can be seen that the five stations XZ, MP, JL, XG, and LJ received abundant lightning electric field pulses from 41.92 to 41.935 s, whereas stations MD and YC received few lightning electric field pulses. The systems of stations MD and YC that received lightning electric field signals were close to saturation, causing the two stations to miss some pulses. This made it impossible to achieve synchronization at all seven stations, which ultimately affected the positioning results. The positioning results during this period are shown in the southeast direction of the channel, which is the negative leader channel of the lightning. This may be because the lightning channel is very close to stations MD and YC, and the pulses of the negative leader are abundant and high power, which causes saturation of the systems of stations MD and YC during this period.

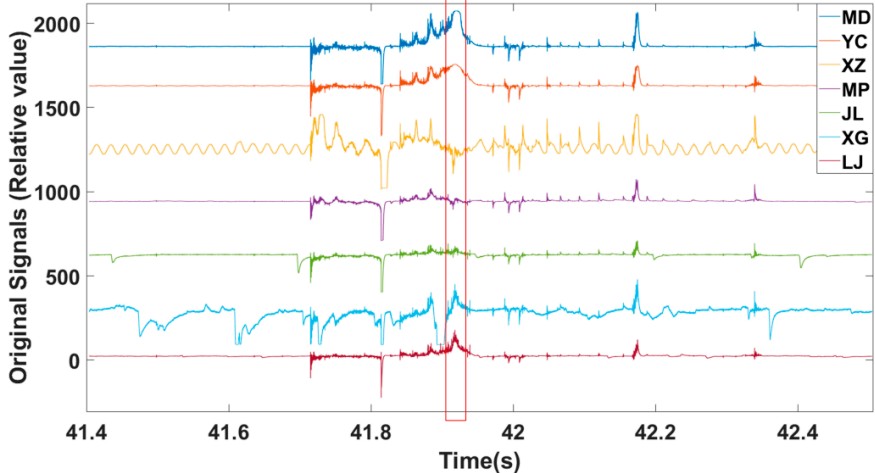

**Figure 13.** Original electric field signal over time from the seven stations for CG flash 000241.

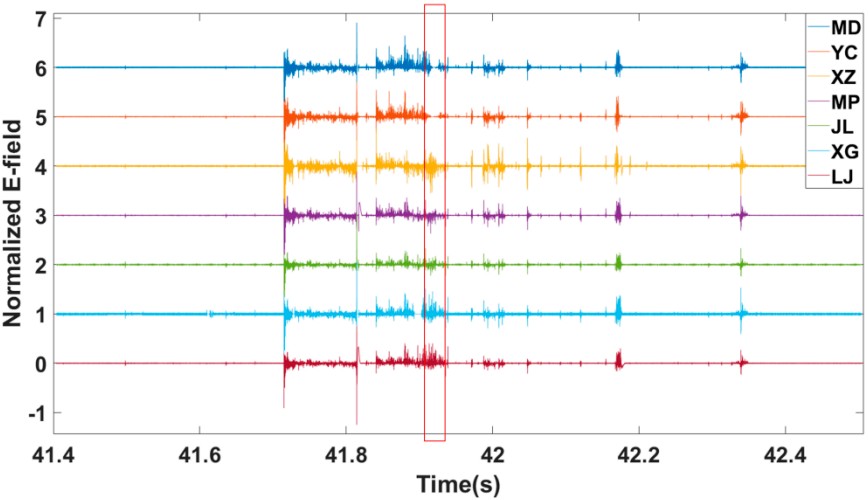

**Figure 14.** Normalized electric field signal over time from the seven stations for CG flash 000241.

## 6. Conclusions

In this paper, Pearson correlation combined with EMD is applied for lightning signal processing and discharge electric field pulse matching, and this method improves the accuracy of pulse matching and the mapping quality of lightning discharges. The Pearson correlation was proven to be a good tool for pulse matching of the same radiation source of lightning. The main conclusions of this paper are as follows.

(1) The lightning electric field signal is decomposed by the EMD method and then partially synthesized, which removes the low-frequency components from the original signal and facilitates subsequent pulse seeking and pulse matching. Normalizing the decomposed and resynthesized signals can make the pulse amplitudes of the same radiation source at different stations more consistent.

(2) After the signal is processed by the MED method, Pearson correlation is applied to match lightning electric field pulses. This paper uses the new method to locate the lightning channels of an IC lightning flash and a CG lightning flash and analyzes the location results for the two lightning flash. Compared with a previous method, the results show that the new method has good performance in lightning location and has significantly improved the accuracy of pulse matching and the mapping quality of lightning discharges.

(3) According to the positioning result of a CG lightning flash, after the three negative recoil streamers were finished (approximately 1 ms, 12 ms, and 2 ms), the negative leader channel underwent a *K*-process. The negative recoil streamers were not connected to the *K*-processes. The three negative recoil streamers of the positive leader channel may have triggered the three *K*-processes of the horizontal negative leader channel of the lightning, respectively.

The preprocessing of lightning radiation signal and the pulse matching of the same radiation source not only has great significance to three-dimensional lightning positioning but also can help with studying the differences in the amplitude and width of electromagnetic pulses from the same radiation source among different stations, which is useful for lightning electromagnetic pulse protection. Obtaining better lightning refined positioning results not only requires improving the performance of data acquisition equipment but also requires further improvements in lightning positioning algorithm. The positioning method of the positive leader radiation source is worthy of further study, which will also be the direction of our next work.

**Author Contributions:** Data acquisition, Y.W.; methodology, Y.W. and Y.M.; software, Y.M. and Y.M.; validation, Y.W., Y.M. and Y.L.; investigation, Y.L. and G.Z.; writing—original draft preparation, Y.W. and Y.M.; writing—review and editing, Y.W., Y.W., Y.L. and G.Z. All authors have read and agreed to the published version of the manuscript.

**Funding:** This work was funded by the National Key Research and Development Program of China (Project Number: 2017YFC1501502), the National Natural Science Foundation of China (Approval Number: 41675006; 41875002), and the Key Laboratory of Middle Atmosphere and Global Environment Observation (Grant/Award LAGEO-2019-06).

**Institutional Review Board Statement:** Not applicable.

**Informed Consent Statement:** Not applicable.

**Data Availability Statement:** The data presented and the code from this paper are available on request from the corresponding author (Wang Yanhui).

**Acknowledgments:** We would also like to thank the Zhang Guangshu Lightning Research Team of the Northwest Institute of Eco-Environmental Resources of the Chinese Academy of Sciences and the Public Technical Service Center of the Northwest Institute of Eco-Environmental Resources of the Chinese Academy of Sciences for their support in the field observations and data acquisition for this article.

**Conflicts of Interest:** The authors declare no conflict of interest.

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
