# Peer review of "A New Approach of 3D Lightning Location Based on Pearson Correlation Combined with Empirical Mode Decomposition"

_remotesensing, doi:10.3390/rs13193883_

Round 1
Reviewer 1 Report
The manuscript presents the application of the method of 'empirical moded decomposition' (EMD) and use of the Pearson correlation coefficient of time series to the location of electrical pulses from lightning in a TOA network. The method is described and an example for its application to 2 lightning flashes is presented and discussed.
The use of the new technique of EMD for the preprocessing of the time series is appropriate for the preprocessing of electrical field time series, since these are affected by noise and also low frequency components. The presented work is therefore of great interest for the lightning location community.
While this paper can give a valuable contribution to the field of lightning location methods it suffers from several issues which are listed in the following.
Major issues/questions:
(1) I appreciate the compactness of the text, however the description of the processing must be given in more detail. This should include the description of the single steps of the method and also the reasons for the method design and the chosen parameters.
Several questions appear:
- Why only the residual component was subtracted from the original data? Is this justified by the special characteristics of the signal? Since the pulses appear as narrow spikes in the time series I could imagine that the sum of the first few (high frequent) IMFs could give even a better pulse (spike) extraction.
- How the 'normalization' (Line 175) was done?
- Particularly the use of the Pearson correlation is hard to understand (lines 190-200). Is it a cross-correlation method for determining the shift between the time series? If yes, then how large is the time window for the pulse matching? (Is it the value of 1500 samples mentioned in Line 208?) This determines the spatial resolution. Do subsequent windows overlap in time?
- How the pulses are identified?
- Is the result the location for a single pulse or for a certain window over the time series?
(2) What is the location uncertainty of the new method for horizontal and for vertical coordinates? This is an important information and can be determined theoretically, by simulations or using observational data.
(3) The comparison of the location results of the old vs. new method (Lines 270-297) shows a significant improving mainly due to a much larger number of located points (matched pulses). The reason for this improvement must be explained. Was it achieved just due to the removal of the residual component? Or was the method of pulse identification and matching procedure improved?
Are the points found with the old method a subset of the new found points?
(4) The discussion and interpretation of the physical content of the 2 location examples in sec. 5.2 is interesting by its own, but it seems unnecessary detailed in this work which is dedicated to the methodical aspect. The same holds for the conclusion (3). Is this a new result which could be obtained only thanks to the new method? If yes, than this fact should be underlined.
Specific issues
- Fig.2-3: The original signal from station XZ shows a persistent medium frequency oscillations. The signal synthesis as described in the text, i.e. removing just the residual component would not remove this oscillation. I suppose there was some additional preprocessing of the XZ time series. Moreover in Fig.3 all the time series have lost any features for times > 37.3 sec.. Was the data cut at this time?
- Line 235: How is a pulse identified? Is there a certain threshold? In fig. 4 there are other spikes (esp. negative) which are not marked as pulses. This information is important, since the 'normalization' was applied.
- Fig.5: Describe what the black dots stand for.
- Fig.4+5: Why not use the same time domain in both figures? This would it make simpler for the reader to compare both figures.
- The time axis information in all the figures should be uniform.
The time format in fig. 7,8,10 is given in 'hour-min-sec', while the other figures show just seconds. How are they related?
- In the Figs. 7,8,10 I suppose the altitude is 'above ground'(?).
The altitude seems to be limited by 7km. Aren't there any signals from higher altitudes? Thunderstorms extend up to more than 12km usually.
- The information stated in Lines 364-368 cannot be found in the presented data on Fig 11.
Author Response
Dear Reviewer:
Thank you for your letter and for the comments concerning our manuscript entitled“A new approach of 3D lightning location based on Pearson correlation combined with empirical mode decomposition”(ID: remotesensing-1359289). Those comments are all valuable and very helpful for revising and improving our paper, as well as the important guiding significance to our researches. We have studied comments carefully and have made correction which we hope meet with approval. Revised portion are marked in yellow in the revised manuscript for comparison. Should you have any questions, please contact us without hesitate. We make our best endeavors to answer each of the questions raised by the reviewers. The detailed responses can be found in the attachment. We first retype these comments in italic font, and then present our response to comments.
We tried our best to improve the manuscript and made some changes in the manuscript. And here we did not list the changes but marked in revised paper.
We appreciate for your warm work earnestly, and hope that the correction will meet with approval. Please see the attachment.
Once again, thank you very much for your comments and suggestions
Yours Sincerely,
Wang Yanhui and Min Yingchang
2021.9.8

Reviewer 2 Report
Dear Authors,
I appreciate your effort in composing the manuscript. I suggest a major review of the presented document. One of the key points is the language, which in its present style makes reading difficult. Half-line sentences are followed by multiline ones relevant to different streams of communication. The goal of every publication is to convey the research in the most transparent and comprehensible way. I urge you think about the reader.
The introduction must be more structured. There are several methods to describe there; take your time to disentangle them into several concise paragraphs. It must be clear how the Pearson correlation method improves the accuracy of pulse matching, and what is also important, what is the advantage in general.
Section 3. The nonlinear least-squares method was introduced long before [10,12].
Section 3.1. Make a short description of the empirical mode decomposition. It starts with a general discussion that abruptly cuts by referencing [8,33]. What is the method, what are the "presets" mentioned? Introduce a couple of equations demonstrating the method. Explain what is meant under the signal synthesis.
Section 3.2. introduce what are the issues in the previous methods, either in the Introduction or here. Explain what A and B stand for in your research. Add several equations for the lines 194-200 so that it is easier to follow. What are the criteria for choosing appropriate n in the present work?
Figures 7 and 8. They would greatly benefit from a mathematical explanation of the presented quantities and what the colors are.
I did not proceed further to avoid irrelevant remarks in the results section. I believe it is also improved when the first sections are edited.
Author Response

(The authors gave the same response as above.)

Reviewer 3 Report
This study seeks to use a combination of correlation and empirical decomposition methods to process VHF source signals from lightning activity. The results of the study show promise in gathering more source data points to analyze per flash as well as lower the uncertainty of the polarity of a lightning flash. Flashes were also geolocated with higher accuracy. Most of the concerns/suggestions/edits are in the enclosed PDF.
An additional question is how well would this technique work in other areas, coastal or marine environment, severe deep convection, weak convection, etc.

Author Response
Dear Reviewer:
Thank you for your letter and for the comments concerning our manuscript entitled“A new approach of 3D lightning location based on Pearson correlation combined with empirical mode decomposition”(ID: remotesensing-1359289). Those comments are all valuable and very helpful for revising and improving our paper, as well as the important guiding significance to our researches. We have studied comments carefully and have made correction which we hope meet with approval. Revised portion are marked in yellow in the revised manuscript for comparison. Should you have any questions, please contact us without hesitate. We make our best endeavors to answer each of the questions raised by the reviewers. The detailed responses can be found in the following. We first retype these comments in italic font, and then present our response to comments.
Responds to the comments:
Thank you very much for your comment. Based on your suggestions, we have made corresponding changes in the text. According to the statistical results of our multiple lightning location data, the error requirement is met when the chi-square value x2<5. Finally, the positioning results are screened by the chi-square value (x2<5). In Section 3 of the revised manuscript (lines 139-140), we described the range of chi-square values used in this paper.
At present, we have not found any application of the Pearson correlation method combined with empirical mode decomposition in coastal or marine environment, severe deep convection, weak convection. However, the EMD data processing method and the Pearson correlation coefficient, as separate methods, appeared relatively early and are currently used in many fields. For example, the Pearson correlation coefficient can be used to express the correlation between lightning frequency and average temperature.
We tried our best to improve the manuscript and made some changes in the manuscript. And here we did not list the changes but marked in revised paper.
We appreciate for your warm work earnestly, and hope that the correction will meet with approval.
Once again, thank you very much for your comments and suggestions
Yours Sincerely,
Wang Yanhui and Min Yingchang
2021.9.8

Round 2
Reviewer 1 Report
I thank the authors for the response to my first review. The added explanations and descriptions improve the manuscript and made it more comprehensive.
In some parts the added text passages even seem too long now (E.g. 167-180 (with the typo 'MED' should be EMD (?) and 215-241) The authors might consider to make these newly included text paragraphs more compact.
2 of the raised issues are still not completely clear for (and I suppose also for the potential reader):
- The normalization is justified (182-190), but still not explained. Please provide a formula. Is it y(t)= x(t)/(max(x)-min(x))*2 -1 ?
or y(t) = x(t)/max(max(x), abs(min(x))) ? - The application of the EMD for preprocessing of the signal: I agree, without doubt the removal of the low frequency components is important and the MED is very appropriate method for this purpose. The authors state that the residual component was subtracted: However: how is this residual component defined/found? In the straight forward EMD the residual component is the remaining monotone function after all IMF were found and subtracted. In the presented work obviously the residual component contains more than just the monotone part. Lines 279-280 list some content of the residual component. However it is not described how this content is identified by the algorithm. Lines 142-167 describe merely the general approach for the EMD. I think it is important to give a short information on the criteria which were used for selecting the IMF and how the signal for the subsequent processing is composed.
Some minor points:
- Lines 252-256 in fact it is a single condition: (2) is just a definition: condition (1) is max(rho)>alpha
- Location accuracy is given in the response. I think it could be useful info for the readers too.
- the authors use the terminus '(small) waveband' I'm not sure if this is the proper wording.
Author Response
Dear Reviewer:
Thank you for your comments concerning our manuscript entitled“A new approach of 3D lightning location based on Pearson correlation combined with empirical mode decomposition”(ID: remotesensing-1359289). Those comments are all valuable and very helpful for revising and improving our paper, as well as the important guiding significance to our researches. We have studied comments carefully and have made correction which we hope meet with approval. Revised portion are marked in yellow in the revised manuscript for comparison. Should you have any questions, please contact us without hesitate. We make our best endeavors to answer each of the questions. The detailed responses can be found in the attachment. Please see attachment.
We appreciate for your warm work earnestly, and hope that the correction will meet with approval.
Once again, thank you very much for your comments and suggestions.
Yours Sincerely,
Wang Yanhui and Min Yingchang
2021.9.19

This manuscript is a resubmission of an earlier submission. The following is a list of the peer review reports and author responses from that submission.